# Case-control investigation of invasive Salmonella disease in Malawi reveals no evidence of environmental or animal transmission of invasive strains, and supports human to human transmission

**Leonard Koolman**[1,2☯], **Reenesh Prakash**[1,2☯], **Yohane Diness**[1], **Chisomo Msefula**[3], **Tonney S. Nyirenda**[3], **Franziska Olgemoeller**[1,4], **Paul Wigley**[2], **Blanca Perez-Sepulveda**[2], **Jay C. D. Hinton**[2], **Siân V. Owen**[2], **Nicholas A. Feasey**[1,4], **Philip M. Ashton**[1,2]*, **Melita A. Gordon**[1,2,3]

1 Malawi-Liverpool Wellcome Programme, Blantyre, Malawi, 2 Institute of Infection, Veterinary & Ecological Sciences, University of Liverpool, Liverpool, United Kingdom, 3 Kamuzu University of Health Sciences, Blantyre, Malawi, 4 Department of Clinical Sciences, Liverpool School of Tropical Medicine, Liverpool, United Kingdom

☯ These authors contributed equally to this work.

* philip.ashton@liverpool.ac.uk

## Abstract

### Background

Invasive Salmonella infections cause significant morbidity and mortality in Sub-Saharan Africa. However, the routes of transmission are uncertain. We conducted a case-control study of index-case and geographically-matched control households in Blantyre, Malawi, sampling Salmonella isolates from index cases, healthy people, animals, and the household environment.

### Methodology

Sixty index cases of human invasive Salmonella infection were recruited (March 2015-Oct 2016). Twenty-eight invasive Non-Typhoidal Salmonella (iNTS) disease and 32 typhoid patients consented to household sampling. Each index-case household was geographically matched to a control household. Extensive microbiological sampling included stool sampling from healthy household members, stool or rectal swabs from household-associated animals and boot-sock sampling of the household environment.

### Findings

1203 samples from 120 households, yielded 43 non-Typhoidal Salmonella (NTS) isolates from 25 households (overall sample positivity 3.6%). In the 28 iNTS patients, disease was caused by 3 STs of Salmonella Typhimurium, mainly ST313. In contrast, the isolates from households spanned 15 sequence types (STs). Two S. Typhimurium isolates from index

**Data Availability Statement:** All sequencing data is available from ENA in the BioProject PRJNA818086: https://www.ebi.ac.uk/ena/browser/view/PRJNA818086. Per sample BioSample accessions are listed in S1 Table.

**Funding:** This project was funded by a knowledge-gap grant from the Bill and Melinda Gates Foundation. The Malawi-Liverpool-Wellcome Programme is core-funded by a grant from Wellcome (206545/Z/17/Z). This work was supported in part by a Wellcome Trust Senior Investigator award [grant number 106914/Z/15/Z] to JCDH. MAG is supported by a Research Professorship from the UK Department of Health National Institute of Health Research (NIHR300039). The funders had no role in study design, data collection and analysis, decision to publish, or preparation of the manuscript.

**Competing interests:** The authors have declared that no competing interests exist.

cases closely matched isolates from their respective asymptomatic household members (2 and 3 SNP differences respectively). Despite the recovery of a diverse range of NTS, there was no overlap between the STs causing iNTS disease with any environmental or animal isolates.

## Conclusions

The finding of NTS strains from index cases that matched household members, coupled with lack of related animal or environmental isolates, supports a hypothesis of human to human transmission of iNTS infections in the household. The breadth of NTS strains found in animals and the household environment demonstrated the robustness of NTS sampling and culture methodology, and suggests a diverse ecology of *Salmonella* in this setting. Healthy typhoid (*S.* Typhi) carrier state was not detected. The lack of *S.* Typhi isolates from the household environment suggests that further methodological development is needed to culture *S.* Typhi from the environment.

### Author summary

Invasive *Salmonella* infections cause the loss of millions of disability adjusted life years (DALYs) every year globally. The two main types of invasive *Salmonella* infections in Africa are i) typhoid fever, caused by *Salmonella* Typhi, and ii) invasive Non-Typhoidal Salmonella (iNTS) disease, primarily caused in our setting by *Salmonella* Typhimurium. Despite the high disease burden, and the observed differences between the epidemiology of typhoid and iNTS disease, we lack an understanding of the reservoirs and transmission routes of iNTS. Therefore, we carried out extensive microbiological sampling of the household members, domestic animals, and living environments of patients with invasive *Salmonella* infections, and of geographically-matched control households, and investigated the genetic relationships between household *Salmonella* and index-case bloodstream isolates by whole genome sequencing (WGS). We identified a wide range of NTS serovars / sequence types across all households and sample-types, but only identified *Salmonella* that matched iNTS that matched invasive cases strains in the stool of healthy people from the same households. Our findings support, but cannot prove, the hypothesis that iNTS-associated strains are transmitted from person-to-person. Boot-sock sampling of the household environment gave the highest yield of *Salmonella* of any of our sampling strategies. None of the 41 environmental *Salmonella* isolates from non-human sources, including 4 domestic animal-associated isolates, matched the disease-causing sequence types. Our findings are consistent with a hypothesis that the reservoir of Typhimurium iNTS infections is the human gastrointestinal tract, and transmission occurs within households. Longitudinal studies are required, however, to confirm this hypothesis.

## Introduction

In High Income Countries (HICs), a range of non-typhoidal Salmonella (NTS) serovars are predominantly associated with enterocolitis and diarrhoeal disease which is often self-limiting, and is associated with zoonotic transmission through industrialised food production. In marked contrast, in Low-Middle Income Country (LMIC) settings *Salmonella* are an

important cause of invasive bloodstream infections (BSI) associated with substantial morbidity and mortality. There are two distinct and contrasting clinical illnesses in the African setting; typhoid fever is a long-recognised illness, caused by the human-restricted pathogen *Salmonella* Typhi, while invasive non-typhoidal *Salmonella* (iNTS) disease is a more recently emerged syndrome in Africa, caused by several NTS serovars that have a wide potential host-range [1,2].

There were an estimated 14.3 million cases of typhoid and paratyphoid fever globally in 2017, resulting in 135,900 deaths, 15.8% of which were in Sub-Saharan Africa [1]. *Salmonella* Typhi is a cause of BSI across the continent, with high incidence and multiple outbreak reports since 2012 [3–7], and Malawi has a very high endemic incidence of 444/100,000 person-years of observation [8].

Globally in 2017, invasive NTS (iNTS) disease was estimated to cause 535,000 illnesses and 77,500 deaths, reflecting its devastating estimated case-fatality of 14.7% [2] [9]. Sub-Saharan Africa (SSA) carries the highest burden disease [10–15], and accounts for 85.8% of global iNTS deaths [2]. While susceptibility to typhoid fever is not associated with underlying conditions, iNTS disease in SSA is particularly associated with immunocompromise, particularly HIV in adults, and malaria, malnutrition or HIV in young children [12,14,16,17].

Two *Salmonella* serotypes, Typhimurium and Enteritidis, are the dominant causes of iNTS across SSA. Data from Malawi, South Africa, Kenya, Mozambique and Mali, show *S*. Typhimurium Sequence Type (ST) 313 (ST313) to be the most common (approximately 80%), and S. Enteritidis ST11 to be the next most frequent (approximately 15%) [17–25].

The transmission routes, sources and reservoirs of these specific strains of NTS responsible for iNTS disease in Africa remain uncertain, but are critical to planning preventive measures. Although they might be transmitted in a zoonotic fashion (analogous with diarrhoeal disease in HICs), previous household or food-chain studies have failed to find any evidence to link the NTS strains responsible for invasive disease with either domestic animals or with food production, while disease case-control studies have established genomic links between isolates from cases and from their healthy household family-members. This raises the possibility that transmission is predominantly human to human rather than zoonotic.

To address the knowledge gap concerning the reservoirs and transmission of NTS Salmonella strains responsible for invasive *Salmonella* infections, we conducted a cross-sectional case-control study following iNTS and typhoid disease, in index-case households and geographically-matched control households across Blantyre, Malawi. This is the first such study to combine simultaneous human, animal, and bootsock household-environmental sampling. Whole genome sequencing of the resulting isolates was used to interrogate at high resolution the relationships between isolates causing invasive disease and isolates from human, animal and environmental household sources.

## Methods

### Ethics

We sought written informed consent from adults, and written consent from the parent or guardian of minors. Ethical permission for this study was granted by the University of Malawi College of Medicine Research Ethics Committee, application number COMREC P.08/14/1617.

### Setting

Queen Elizabeth Central Hospital (QECH) provides free healthcare to the approximately 1.3 million inhabitants of Blantyre District, Malawi, and is the tertiary referral hospital for the southern region of Malawi. Since 1998, the Malawi-Liverpool-Wellcome Trust Clinical

Research Programme has provided a quality assured diagnostic blood culture service for febrile adult and paediatric medical patients admitted to QECH. This service is provided for admitted adults (>16 years old) with axillary temperature over 37.5˚C or clinical suspicion of sepsis, and for children (<16 years old) who were malaria slide negative, or positive and critically ill, or with clinical suspicion of sepsis. Maps were obtained from https://data.humdata.org/dataset/cod-ab-mwi, with the exact link https://data.humdata.org/dataset/20eb8e5b-134d-41d8-a56f-4f358f7faf16/resource/50f185b1-b028-4787-a591-80c8db81cfed/download/mwi_adm_nso_20181016_shp.zip. The map shape files are licenced under the Creative Commons Attribution for Intergovernmental Organisations licence - https://creativecommons.org/licenses/by/3.0/igo/legalcode.

### Case and control recruitment

The first two eligible cases of blood culture-confirmed adult or paediatric invasive *Salmonella* infection (one each of iNTS and typhoid fever) presenting each week at QECH during two periods of recruitment (February to May 2015 and November 2015 to October 2016) were recruited. On 2 weeks when no iNTS case presented, it was replaced with a second case of typhoid. Patients living outside the Blantyre district, and those with recurrent iNTS disease (a known second blood culture positive episode in the previous 30 days) were excluded. Following recruitment, the field team visited the index cases in their households, where GPS co-ordinates were taken and a household socio-demographic questionnaire was completed. Index case households were sampled within a maximum of 14 days following initial presentation of the index case to QECH. Control households were then selected by random bottle-spin and pacing 100 m from the index case household, recruited with informed consent, and GPS and questionnaire data were also collected. Exclusion criteria for individual household members were current treatment for recent invasive *Salmonella* disease, or declining consent. In the event of exclusion criteria being met, the next-nearest house in the same direction was selected. Household members were defined as people who usually sleep in the household.

### Sampling methodology

We carried out a microbiological survey of the index-case household and the control household, comprising stool from household members, stool or rectal swabs from domestic and household-associated animals and systematic boot-sock sampling of the living environment (latrine, rubbish area, bedroom, cooking area and the house perimeter).

*Stool (humans)*: Sample containers were provided 24–48 hours before the main sample-collection visit, and we requested that up to three consecutive stools were collected from each individual in the household who was available during that time period. The three samples belonging to each person were pooled, to maximise recovery of *Salmonella*.

*Stool (animals)*: Fresh stool or rectal swab samples were collected from domestic or household-associated animals (cows, chickens, goats, pigs, dogs, cats, rats, doves, guinea pigs or gecko lizards).

*Environmental*: The investigator's shoe was first covered with a waterproof sterile bag to prevent cross-contamination, then a sterile bootsock was put over this, moistened with sterile water, and single bootsocks were used to collect samples from each of the follow sites: pit-latrine, outdoor rubbish area, cooking and bedroom areas, and from the outside perimeter of the household, using a standardised protocol for location and number of paces (10 paces), and a step and twist motion. Bootsocks were then removed and placed into a sterile plastic bag for transport.

All samples were transferred from the field to the laboratory in cool boxes and processed on the same day.

## Sample enrichment and culture

S1 Fig shows a schematic of sample processing. A portion (~1g) of stool was emulsified in 9ml of Selenite F selective enrichment broth (Oxoid, UK) and incubated at 37˚C for 18–24 hours. For samples other than stool (i.e. bootsocks, swabs), we used a pre-enrichment step; these samples were incubated in Buffered Peptone Water (BPW) at 37˚C for 18–24 hours. One ml of the BPW sample was then transferred into 9 ml Selenite broth and incubated for a further 18–24 hours at 37˚C.

After incubation, 100µl of bacterial culture taken from the surface of the Selenite broth was sub-cultured onto Xylose Lysine Deoxycholate (XLD) agar, incubated at 37˚C for 18–24 hours and examined for suspected colonies of *Salmonella* (red/pink colonies with black centres). Pure candidate colonies were biochemically confirmed as *Salmonella* species using API 20E (BioMerieux). Putative *Salmonella* isolates were sub-cultured onto Nutrient agar (Oxoid, UK) and incubated for 24 hours at 37˚C. Following growth, a drop of 0.85% saline was placed onto a microscope slide and a single colony from the Nutrient agar was emulsified into the saline. Serotyping was performed by the slide agglutination technique. Positive serotyping was confirmed if agglutination occurred in antigens O4, O9, Vi, Hd, Hg, Hi, Hm. If an organism was positive for Vi antigen but negative for O9 antigen, a dense suspension of the organism was prepared in 0.85% saline, autoclaved at 121˚C for 15 minutes and the agglutination repeated. *S.* Typhi was differentiated from NTS isolates if agglutination occurred for O9, Vi and Hd antigens. All API 20E confirmed *Salmonella* isolates which were not *S.* Typhi were sent to the University of Liverpool, UK, for WGS.

## Whole Genome Sequencing (WGS)

Bacterial genomic DNA was extracted from overnight cultures in LB (1% tryptone, 0.5% yeast extract, 0.5% NaCl; pH 7.0) using the DNeasy Blood and Tissue kit (Qiagen), as per the Gram-negative bacteria protocol for the Quick-DNA Universal Kit (Zymo Research), Biological Fluids & Cells protocol. Sequencing libraries were constructed using the TruSeq DNA PCR-free library preparation kit (Illumina) or the TruSeq Nano DNA HT library preparation kit (Illumina) using a target insert size of 550bp. Libraries were sequenced on an Illumina MiSeq instrument using a 2x250bp paired end protocol at the Centre for Genomic Research, University of Liverpool, UK. Raw sequencing reads were trimmed to remove Illumina adapter sequences using Cutadapt [26] and further trimmed to remove poor quality sequence using Sickle [27] version 1.2 with a minimum window quality score of 20, discarding reads which were less than 10 bp after trimming. Trimmed reads were submitted to Enterobase (https://enterobase.warwick.ac.uk/) for draft genome assembly, MLST assignment, *in silico* serotype prediction and phylogenetic analysis [28]. SNP identification within Enterobase was carried out via a mapping and SNP calling pipeline, and phylogenetic tree construction done with RAxML [29–32] Determinants of antimicrobial resistance genes were identified using the amr-finder-plus software version 3.9.8, using database version 2020-12-17.1 [33].

## Results

Sixty-seven index-case households (i.e. the households of people diagnosed with invasive Salmonelloses) associated with 35 typhoid and 32 iNTS patients were eligible for enrolment between March 2015 and October 2016. Seven index-case households declined to participate, leaving 32 typhoid index-case households with 32 paired control households, and 28 iNTS index case households, with 28 paired control households recruited for the study, giving a total

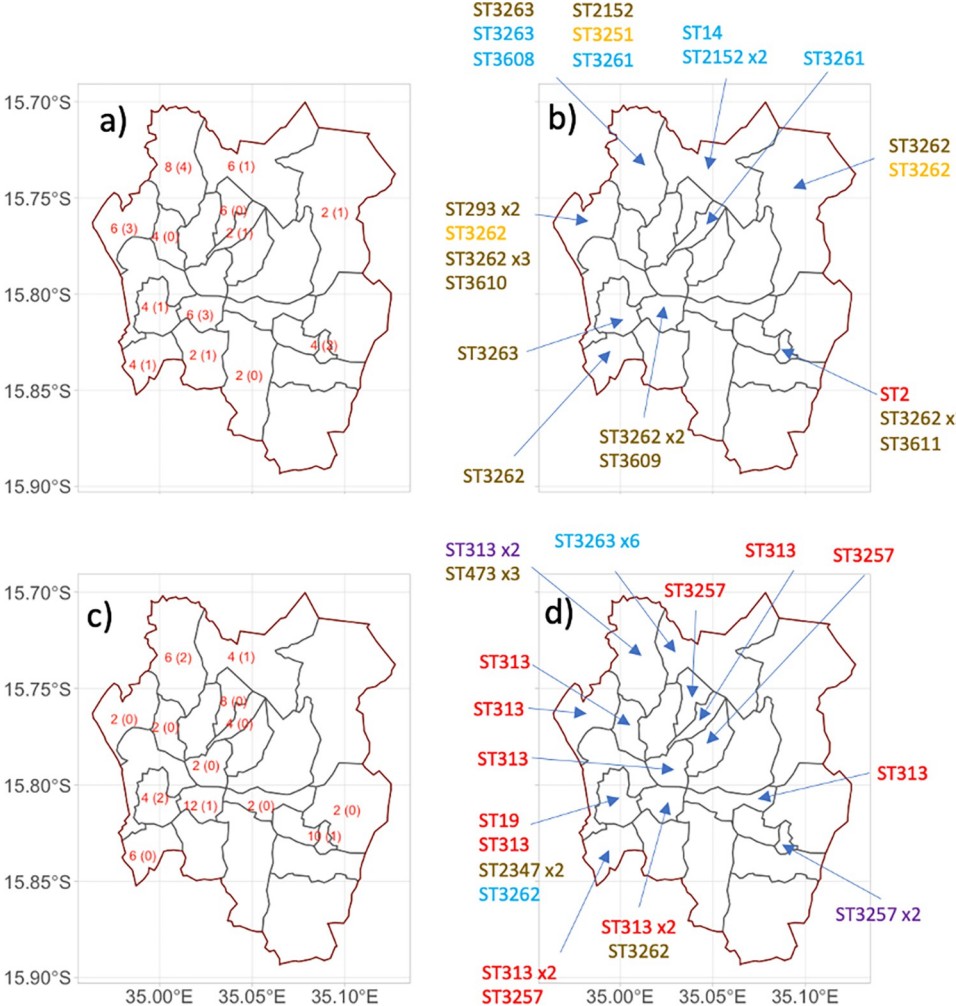

**Fig 1.** The geographic location of samples in this study a) & b) are for typhoid fever case and control households, while c) and d) are for iNTS case and control households. a) and c) indicate the number of households (case + control) that were sampled from each area, in parentheses is the number of households where Salmonella was isolated (not including the index case isolate). b) and d) indicated the sequence type of all the *Salmonella* isolated from that location, including the index case isolates. STs are coloured by the type of sample they were isolated from. Red is invasive Salmonella, blue is a household member of an invasive Salmonella case, purple is in both invasive Salmonella and a household member, brown is the environment and orange is animal. Map shapefile obtained from https://data. humdata.org/dataset/20eb8e5b-134d-41d8-a56f-4f358f7faf16/resource/50f185b1-b028-4787-a591-80c8db81cfed/ download/mwi_adm_nso_20181016_shp.zip. The map shape files are licenced under the Creative Commons Attribution for Intergovernmental Organisations licence - https://creativecommons.org/licenses/by/3.0/igo/legalcode.

of 120 households (for geographic distribution of cases and controls, see Fig 1A). All households were in high-density low-income housing locations within Blantyre. No cases of recurrent iNTS disease were excluded, and no household members were excluded because of current antibiotic treatment for iNTS disease. Table 1 summarises demographic details and sampling across the study for cases and controls.

## Isolation of Salmonella by culture

*Salmonella* bacteria (all NTS) were isolated from 25/120 (21%) households, however, no *S.* Typhi were isolated from any category of household samples, including healthy human family

**Table 1. Summarised demographic information about cases, and case and control households.**

|  | Cases | Controls |
|---|---|---|
| Median age of index case, years (IQR) | 5 (3–10) | NA* |
| Number of male index cases (percentage of total) | 25 (48%) | NA* |
| Median number of people in household (IQR) | 5 (4–6) | 5 (3–6) |
| Percentage of household under 15 years old (IQR) | 42% (33–50%) | 50% (33–53%) |
| Proportion of stool samples from people under 15 years old | 53% | 48% |

* NA = non-applicable.

members. We collected a total of 1203 samples from the 120 households (a mean of 10 samples per household), yielding 43 isolates (all NTS) which were confirmed as *Salmonella* by WGS, giving an overall sample positivity of 3.6% (Table 2). Overall, sample positivity was similar for human, animal and bootsock environment samples (human stool 16/491, 3.3%; animals 4/110, 3.6%; bootsocks 23/620, 3.8%), and across different household categories i.e. iNTS vs typhoid fever, case vs control (Table 2). Consistent sample positivity rates suggest that sampling methods were robust. Animal sampling was from 110 animals in 71/120 (59%) households, and included both domesticated and non-domesticated species (cows, chickens, pigs, dogs, cats, rats, doves, guinea pigs and geckos). The reason for non-sampling from some households was non-ownership of domestic animals.

## Relatedness of sequence types from different sample categories

We isolated a wide wide diversity and relationships of Sequence Types (STs) from different sample categories (invasive disease, healthy human stool, animals, and the household environment) (Fig 1 and Table 3, full line list in S1 Table). Among invasive NTS disease cases, 8 of 27 index case NTS isolates could not be resuscitated from the freezer archive and so could not be genome-sequenced. The remaining 19 typed index NTS isolates were all *S*. Typhimurium and included just 3 STs; 13 were ST313, 5 were ST3257 and 1 was ST19. ST3257 varies from ST313 by only a single MLST locus. One representative typhoid index strain (ST2) was also included

**Table 2. Sources of *Salmonella* isolates from case and paired control households linked to iNTS and typhoid patients.** HH stands for household, and an index case refers a diagnosed case of invasive *Salmonellosis*.

|  | iNTS Index Case Households (HH) (n = 28) | | | | | | iNTS Control Households (HH) (n = 28) | | | | | |
|---|---|---|---|---|---|---|---|---|---|---|---|---|
|  | Total samples | No. samples with Salmonella | % positive samples | No. HH tested | No. HH with Salmonella | % Positive HH | Total samples | No. samples with Salmonella | % positive samples | No. HH tested | No. HH with Salmonella | % Positive HH |
| Human Stool | 116 | 9 | 7.8 | 28 | 4 | 14.3 | 97 | 0 | 0.0 | 28 | 0 | 0.0 |
| Animal stool/swab | 27 | 0 | 0.0 | 20 | 0 | 0.0 | 38 | 0 | 0.0 | 17 | 0 | 0.0 |
| HH environment | 137 | 0 | 0.0 | 28 | 0 | 0.0 | 131 | 6 | 4.6 | 28 | 3 | 10.7 |
| **Total** | **280** | **9** | **3.2** | **28.0** | **4** | 14.3 | **266** | **6** | **2.3** | **28** | **3** | **10.7** |
|  | Typhoid Index Case households (HH) (n = 32) | | | | | | Typhoid Control households (HH) (n = 32) | | | | | |
| Human Stool | 157 | 4 | 2.5 | 32 | 2 | 6.3 | 121 | 3 | 2.5 | 32 | 3 | 9.4 |
| Animal stool/swab | 22 | 2 | 9.1 | 18 | 2 | 11.1 | 23 | 2 | 8.7 | 16 | 2 | 12.5 |
| HH environment | 168 | 5 | 3.0 | 32 | 4 | 12.5 | 166 | 12 | 7.2 | 32 | 9 | 28.1 |
| Total | **347** | **11** | **3.2** | **32** | **7** | **21.9** | **310** | **17** | **5.5** | **32** | **11** | **34.4** |

**Table 3. The number of isolates belonging to each ST identified in this study, from each niche.**

| ST | Animal | Boot sock | Family member stool | Index case | Grand Total |
|---|---|---|---|---|---|
| 2 | | | | 1 | 1 |
| 14 | | | 1 | | 1 |
| 19 | | | | 1 | 1 |
| 293 | | 2 | | | 2 |
| 313 | | | 1 | 13 | 14 |
| 316 | 1 | | | | 1 |
| 473 | | 3 | | | 3 |
| 2152 | | 1 | 2 | | 3 |
| 2347 | | 2 | | | 2 |
| 3257 | | | 1 | 5 | 6 |
| 3261 | 1 | | 2 | | 3 |
| 3262 | 2 | 10 | 1 | | 13 |
| 3263 | | 2 | 7 | | 9 |
| 3608 | | | 1 | | 1 |
| 3609 | | 1 | | | 1 |
| 3610 | | 1 | | | 1 |
| 3611 | | 1 | | | 1 |
| **Grand Total** | **4** | **23** | **16** | **20** | **63** |

for reference. Among healthy household members, 16 isolates represented 8 STs; among animals there were 4 isolates from 3 STs; and from environmental samples there were 23 isolates from 9 different STs.

Of the 19 index iNTS case isolates from blood culture, 19/19 were *S.* Typhimurium, of 3 different STs; ST313 (n = 13), ST3257 (a Single Locus Variant of ST313, n = 5) and ST19 (n = 1) (S1 Table and Fig 2). There were two instances of the same sequence type of S. Typhimurium being isolated from both invasive index-case isolates and healthy human household samples. Specifically, 2 cases of invasive disease caused by *S.* Typhimurium ST313 and ST3257 were linked to isolates from 2 healthy household members who shared the same address and also had ST313 and ST3257 respectively isolated from their stools (Fig 2). In both cases the household member carrying a matching isolate was an adult, whereas stool isolates of *Salmonella* from household members were found equally among adults (n = 7) and children (n = 8).

Fig 2 demonstrates the relationships of all sequence types and sample categories. Fig 1 additionally demonstrates the geographical spread of sampling and of isolation of Salmonellae across Blantyre, and the geographical distribution of all STs isolates across different wards of the city. We found no overlap in STs between isolates causing invasive human disease with any animal or environmental isolates, despite evenly-spread geographical sampling and a well distributed and consistent rate of isolation from all 3 sample-types. We did identify non-co-localised overlap between some STs from healthy humans and household-associated animals. A monophasic ST3262 was found in a cat, a gecko, and a human, but in geographically distinct households (see Fig 1 map and S1 Table). Similarly, ST3261 (serovar Agoueve/Cubana) was isolated from 2 healthy humans and a chicken, but these were in different households (Table 2). ST316 (serovar Montevideo) was found only in an animal (gecko).

ST/serovars (as identified *in silico*) that were carried by healthy human family members were ST3263 (Havana, n = 7), ST313 (Typhimurium, n = 2), ST2152 (Gaminara, n = 2), ST3261 (Agoueve/Cubana, n = 2), ST14 (Seftenberg, n = 1) and ST3608 (Ogbete, n = 1). The wide range of ST/serovars found across the household environment included monophasic

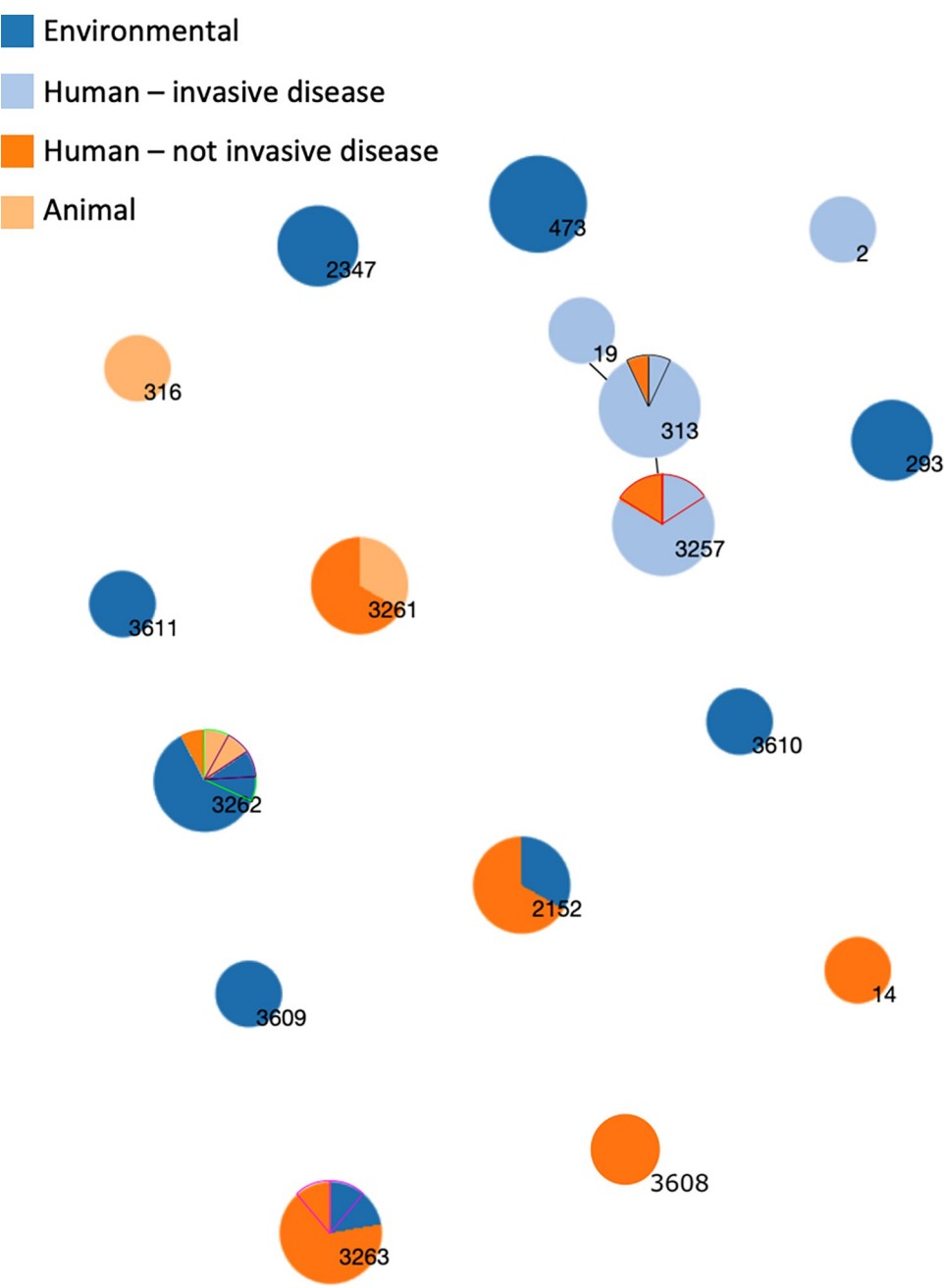

**Fig 2. Minimum spanning tree based on seven MLST genes showing the genetic similarity between different niches–the only non-clinical niche where STs associated with human invasive disease were present was the gut of household contacts.** Each circle represents a sequence type, the size of the circle is proportional to the number of isolates of that ST. STs that vary by one locus are joined by a black bar. Two invasive disease index isolates had a matched isolate from a healthy human sample of the same ST from the same household. These isolates are indicated by black and red outlines. In cases where there were multiple isolates of the same sequence type isolated from the same household, these "slices" of pie have had a border of the same colour. I.e. There were two isolates of ST3257 isolated from the same household, one from human invasive disease, and one from human nont invasive disease. Therefore, on the ST3257 pie, one slice of orange (not-invasive) and one slice of light blue (invasive) have been bordered in red.

II42:r:-|IIIb 42:r:-:[z50] (10), ST473 (Hadar n = 3), ST3263 (Havana, n = 3), ST293 (Amager, n = 2), ST3347 (Mgulani, n = 2) ST3609 (Djama, n = 1) and ST3610 (Aberdeen, n = 1).

Some STs isolated from the environment overlapped with those from healthy humans, and household animals. Of 9 STs found in the environment, 3 (ST3262 monophasic; ST3263 Havana; ST2152 Gaminara) showed some overlap with healthy human or animal isolates. However, a further 6 STs were found only in the household environment, revealing the diversity of *Salmonella* strains in this setting.

There was a wide diversity of ST/serovars, and many STs/serovars were represented by just a single isolate. Where there were multiple isolations of the same serovar, however, a large proportion were co-localised within the same household. Striking examples of this are: of 9 isolates of ST 3263 (*S*. Havana), 6 arose from a single household (9S), representing 6 out of 7 children sampled; of 3 isolates of ST 473 (*S*. Hadar), all arose from the same household (25C, latrine, bedroom and cooking area); of 12 isolates of monophasic II42:r:-|IIIb 42:r:-:[z50] there were multiple positive samples in three households, originating from both animal and environmental samples in the same households. Similarly, 2 isolates each of ST23347 (*S*. Mgulani) and ST 293 (*S*. Amager) isolates were isolated only from the same respective households (16C and 56C). Thus the data indicated a very marked geographical diversity between households, and a degree of homogeneity within households among NTS that were not linked to invasive disease.

## Whole genome phylogeny of index/household pairs

We used whole genome-derived phylogenetics to infer the high-resolution relationship between the human invasive disease cases of *S*. Typhimurium ST313 & ST3257 and the stool carriage isolates of their respective index household members. In both the ST313 and ST3257 pairs, the isolate that caused the invasive disease formed a monophyletic group with the carriage isolate from the same index household (Fig 3). The SNP distance was 2 SNPs in one case, and 3 SNPs in the other (Fig 3).

A summary of antimicrobial resistance genes present from WGS data for 63 isolates from index cases of human invasive disease and household sampling is available in S1 Table.

## Discussion

Our findings support the overarching hypothesis that iNTS disease in sub-Saharan Africa is transmitted from person to person. We studied 60 index cases of invasive Salmonella disease (32 typhoid and 28 iNTS disease), and took 1203 samples from 120 index and geographically-matched control households, including samples from 110 animals and 491 family members. Nineteen bloodstream isolates sequenced from iNTS cases were of just 3 STs of *S*. Typhimurium. 43 NTS strains were isolated, including 16 from the stool of healthy household members, of which two closely matched the strains from their corresponding household invasive index-case, being within a 2–3 SNP distance from the index case isolate. None of the 27 other *Salmonella* isolates from the environment and animals could be linked to iNTS case isolates.

The reservoirs and transmission route of the pathogens responsible for iNTS in SSA remains unknown. Seasonal patterns of iNTS, and the close domestic association of humans and animals in parts of SSA might suggest that environmental factors could play a role [34,35]. However, numerous studies have failed to establish any convincing link between strains causing invasive disease, with domestic or household animals or with meat-production. Serotypes associated with iNTS have been isolated from animals in South Africa, but these lacked higher-resolution genetic characterisation at sequence level [36,37]. An early study from Kenya using Pulsed field Gel Electrophoresis (PFGE) also found that iNTS-linked serotypes were only

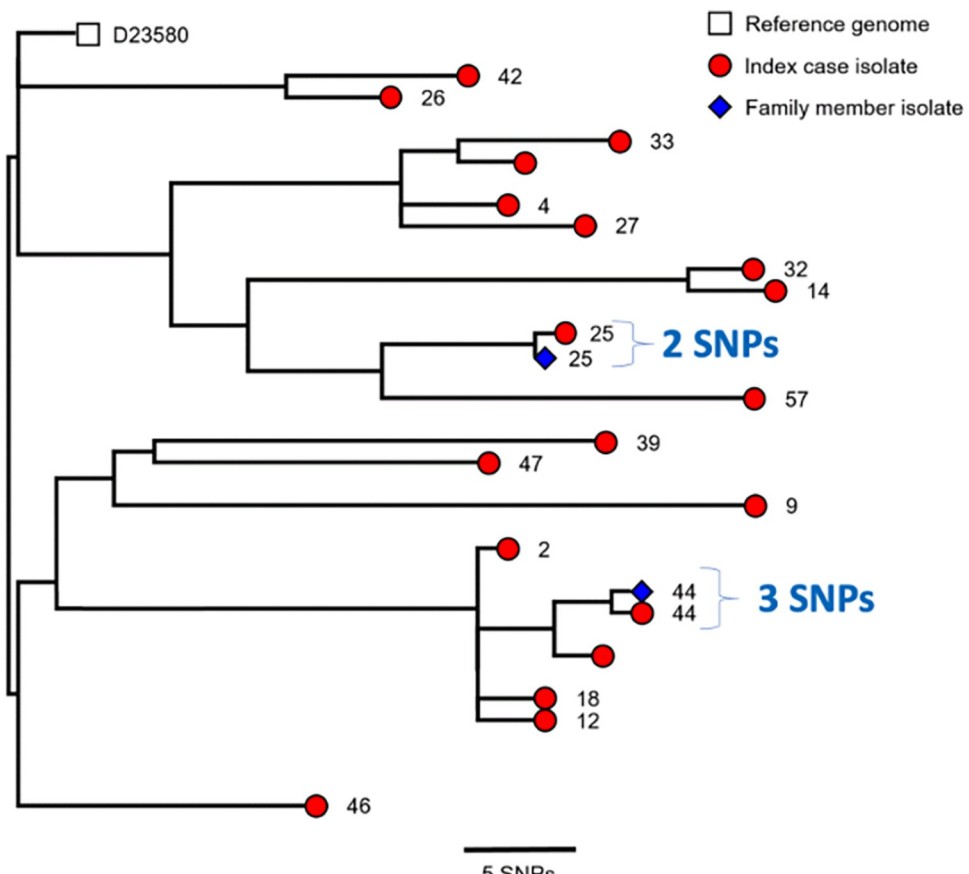

**Fig 3. The close relationship between the index case and family member isolates from households 25 and 44 shown with a phylogenetic tree.** Whole genome maximum likelihood tree of 19 S. Typhimurium isolates described in Table 2. The numbers at the end of each branch indicate household, as described in the first column of Table 2 (all these households are case households). Sequencing read sets were compared against the reference genome D23580 as described in Methods.

rarely found in environmental samples such as water from nearby rivers, household and market vendor food, or from household animals, but PFGE is a low resolution method that is now superseded by Whole Genome Sequencing (WGS) [38]. More recently, iNTS infections caused by *Salmonella* Typhimurium or Enteritidis in an informal urban settlement in Kenya were not found to be epidemiologically associated with the rearing of any domestic animals [39]. In The Gambia, no overlap was observed between STs from iNTS cases and those from the rectal swabs from domestic animals in the households of cases [40], while in Tanzania and Kenya, extensive sampling of "meat pathway" isolates and human disease isolates only isolated ST313 from human samples [41]. Although ST313 was isolated from pigs in Nairobi, this was genomically related to an ST313 variant from the UK that is not associated with invasive disease [42,43].

In contrast, in keeping with our study, several previous studies have provided evidence of a link between strains causing a case of iNTS disease and strains from their healthy family members. In a recent study in Burkina Faso of the households of 29 iNTS cases, 500 samples from humans, domestic animals and water yielded 34 Salmonella isolates of a diversity of serovars. No matched-control households were studied. The only isolates that matched corresponding index cases were obtained from the stool of 3 household members, which corresponded to 4

index cases (2 cases from one house). Salmonella was not isolated from any water sample, and environmental bootsock samples were not taken [44]. Recent microbiological investigations in Kenya, based on whole genome sequencing, found isolates in the stool of healthy age-matched control children that were genomically very closely related to index cases of iNTS disease in the same community. These children were not, however, co-located in the same households [45].

The Global Enteric Multicentre Study of moderate-severe diarrhoea also found strains of invasive disease-related NTS in the stool of 42 children with diarrhoea and 17 healthy control children, highlighting the potential role of the human gut as a source of iNTS infections, but these cases were not directly linked to cases of invasive disease [46].

The findings in relation to *S*. Typhi the human-restricted causative agent of typhoid fever in Africa, provide an interesting epidemiological and biological contrast to our findings for iNTS disease. Although asymptomatic human carriage is known to be one source of transmission of typhoid, we found no evidence of this in any household, including the households of cases. This is unsurprising since a much larger community-based study in a population of approx 100,000 individuals in Blantyre also found no evidence of asymptomatic carriage in any age-group [8]. Since *S*. Typhi is known to be human-restricted, it is likely that transmission of typhoid in Malawi is via stool shedding from acute cases. *S*. Typhi is believed to be transmitted by poor sanitation and contaminated water, entering a unique viable but non-culturable state outside the human host, making it unsurprising that we did not isolate this organism from an of our household samples. Delineating the relative contributions of long-cycle transmission (i.e. waterborne or environmentally mediated) and short cycle transmission (i.e. within the household) has remained difficult for typhoid, for these specific biological and technical reasons [47,48]. It is, however, notable that there was a high rate of isolation of NTS from all sample-types in both typhoid case and control households.

This study also provides some wider insights into household and environmental niches for NTS. Whilst previous case-control studies have focused on domestic animals, adults and children in the same household, here we have additionally studied the household environment. This yielded a similar culture positivity-rate to other sample-types, suggesting that our culture methods were robust, and environmental samples yielded a wide range of ST/serovar strains. Salmonellae appear to have a wider range of relatively unique niches in the household environment than has been previously appreciated in this setting. Although we identified a diversity of serovars, the complete lack of overlap of strains causing invasive disease with either animal or environmental strains was very striking. Overall, there was also a surprising lack of mutual overlap between the ecological niches of serovars colonising the household lived-in environment, health humans and animals. This suggests that NTS strains in general are relatively adapted to their own niches, and this is perhaps in keeping with the very narrow range of STs that are known to cause human invasive disease [49]. Where overlap in strains between healthy humans, animals and the environment did occur, it appeared to be mostly localised to a single household, or a few households.

This study has several important strengths. It is the first to use a case-control approach, and the first to link index case isolates with sampling from such a wide range of simultaneous sources—healthy humans, domestic animals and the lived-in household environment. Sampling was geographically well distributed across all areas of the city. The average household size in Blanyre is 4.3, with 48% of the population under the age of 18 [8]. Based on this, we would predict 528 "available" people in 60 households, and we actually obtained 491 samples, suggesting that 93% of predicted total possible samples were collected (range 81%-113% of predicted in each category), indicating good overall coverage of household members. From the same census, we would expect 48% of the community to be aged less than 18, and the actual

percentage of samples from children was 46.6%, again suggesting highly representative coverage of sampling between adults and children. Yield was similar across all sample-categories, and similar across all areas of the city, suggesting that our culture methods were robust and consistent.

Our methodology was able to identify microbiological connections between people and their household environments at both the individual household level and the community level. The presence of the same STs in multiple samples from the same household indicates that our methodology can resolve this connectivity when it is present in this setting, and this strengthens the significance the "negative" finding of the absence of a link between *Salmonella* causing iNTS disease and animals or the environment. This excellent yield from bootsocks for detecting *Salmonella* in other contexts has been reported elsewhere [50], and substantially enriched the diversity of data and environmental resolution in our study. We recommend that similar studies carried out in the future also use this method.

There are also some limitations to our study. Both the index case and the ST313 colonised household members could have been simultaneously infected from an unsampled environmental reservoir or source. In addition, there was a lag-time of a maximum of 14 days between the presentation of the index case and investigation of the household, which could mean that a common source might be missed. However, this study provides the most comprehensive results from a wide range of simultaneous potential household sources yet to be published, but despite consistent positive yields from all categories of samples we were still unable to find evidence of a "common source". If the index-case were infected outside the home e.g. at work, school, or nursery, which is a possible risk factor in our setting [47], then our study would not identify that microbiological source. Another limitation of our sampling is that it is possible that some household members were on antibiotics at the moment of sampling, additionally, we did not collect information on the precise number of bowel motions sampled by each participant, which could lead to differences in yield between participants.

In addition, an intrinsic limitation of a cross sectional study is that it is not possible to definitely demonstrate the transmission of isolates between reservoirs, sources and cases, and a more comprehensive longitudinal study would be required to prove a transmission route. The accumulated body of "negative" evidence from this and previous cross-sectional studies may now justify the greater resources that would be required to demonstrate "positive" proof of transmission. We had hoped to sample food, but we found that in practice most households lacked refrigeration, and there was rarely food in the house to sample, as meals were eaten at one sitting, so this objective was not realistic in this study. Although previous household studies have not achieved yield of NTS from household water, drinking and washing/cleaning water-sources could be involved in the transmission of organisms causing iNTS, as is seen for *S*. Typhi [47,51], but further methodological development to filter large volumes would be required for these studies.

## Conclusions

In summary, we conducted comprehensive environmental and contact sampling within the households of patients with invasive *Salmonella* disease, and geographically matched control households. Despite comparable yield from all sample types, the only household isolates that matched iNTS cases came from the stool of household members. This study contributes to accumulating evidence that the reservoir of iNTS infections in Africa is likely to be the human gastrointestinal tract. Significantly, no strains associated with human iNTS disease were identified among any household-associated animals, despite extensive sampling and consistent recovery from animal samples. The use of bootsocks to sample the lived-in household

environment uncovered a much wider range of NTS isolates that were not associated with invasive disease, across all sampling sites, than has previously been appreciated. This is consistent with a diverse ecology for *Salmonella* in the household environment, with multiple ST/serovars apparently occupying relatively distinct niches.

## Supporting information

**S1 Table. Salmonella isolates sequenced as part of this study. HH stands for household.**
(XLSX)

**S1 Fig. Flow diagram of sample processing.**
(PPTX)

## Acknowledgments

We are grateful to Alistair Darby, John Kenny and staff at the Centre for Genomic Research at the University of Liverpool for assistance with genome sequencing.

## Author Contributions

**Conceptualization:** Jay C. D. Hinton, Nicholas A. Feasey, Melita A. Gordon.

**Data curation:** Philip M. Ashton.

**Formal analysis:** Philip M. Ashton.

**Funding acquisition:** Melita A. Gordon.

**Investigation:** Leonard Koolman, Reenesh Prakash, Yohane Diness, Chisomo Msefula, Franziska Olgemoeller, Blanca Perez-Sepulveda, Siân V. Owen, Nicholas A. Feasey, Philip M. Ashton, Melita A. Gordon.

**Methodology:** Leonard Koolman, Reenesh Prakash, Chisomo Msefula, Tonney S. Nyirenda, Franziska Olgemoeller, Paul Wigley, Blanca Perez-Sepulveda, Nicholas A. Feasey, Melita A. Gordon.

**Resources:** Jay C. D. Hinton, Melita A. Gordon.

**Supervision:** Jay C. D. Hinton, Melita A. Gordon.

**Writing – original draft:** Leonard Koolman, Reenesh Prakash, Philip M. Ashton, Melita A. Gordon.

**Writing – review & editing:** Leonard Koolman, Reenesh Prakash, Yohane Diness, Chisomo Msefula, Tonney S. Nyirenda, Franziska Olgemoeller, Paul Wigley, Blanca Perez-Sepulveda, Jay C. D. Hinton, Siân V. Owen, Nicholas A. Feasey, Philip M. Ashton, Melita A. Gordon.

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
