## [Decision Letter · Decision Letter 0]

10 May 2022

Dear Dr Ashton,

Thank you very much for submitting your manuscript "Case-control investigation of invasive Salmonella disease in Africa reveals no evidence of environmental or animal reservoirs of invasive strains" for consideration at PLOS Neglected Tropical Diseases. As with all papers reviewed by the journal, your manuscript was reviewed by members of the editorial board and by several independent reviewers. In light of the reviews (below this email), we would like to invite the resubmission of a significantly-revised version that takes into account the reviewers' comments. 

We cannot make any decision about publication until we have seen the revised manuscript and your response to the reviewers' comments. Your revised manuscript is also likely to be sent to reviewers for further evaluation.

Sincerely,

Andrew S. Azman

Deputy Editor

Reviewer's Responses to Questions

**Key Review Criteria Required for Acceptance?**

**Methods**

-Are the objectives of the study clearly articulated with a clear testable hypothesis stated?

-Is the study design appropriate to address the stated objectives?

-Is the population clearly described and appropriate for the hypothesis being tested?

-Is the sample size sufficient to ensure adequate power to address the hypothesis being tested?

-Were correct statistical analysis used to support conclusions?

-Are there concerns about ethical or regulatory requirements being met?

Reviewer #1: Methods: 

- Line 145-146: Are predefined criteria for suspicion of sepsis used? 

- Line 152: "with recurring iNTS disease were excluded". Do you mean deduplication or first presentation also excluded?

- Line 155: Sociodemographic data were collected. Please provide some basic sociodemographic data here (including household size and age of household members). 

- Line 157: GPS coordinates were collected. To allow insights in the geographical distribution of households and geographical interrelatedness of isolates, I strongly suggest to provide a map displaying all sampled households and isolation of iNTS.

- Line 158-159: "Exclusion criteria for control households were current treatment for invasive Salmonella disease for any family member" Were there any household members from index cases who were receiving antibiotics? 

Also exclusion if antibiotic treatment ongoing for an illness not caused by Salmonella? 

- Line 169: "Available household members". Please be more specific: at home during the visit? what was the coverage: how many household members out of all household members were sampled? was there a difference in coverage between adults and children? 

- Line 199: Suggestion to move sentence "Serotyping was performed by the slide agglutination technique." to line 203 (after discussion of subculturing).

- Supplementary Figure 1 does not specify the place of serotyping in the laboratory work-up. 

- Line 224: All serotype data in the paper were inferred from WGS and phenotypic serotyping only used to decide whether or not the isolate should be shipped to the UK (line 209)? Please clarify.

Reviewer #2: The objective of this study—to examine potential household sources of invasive non-typhoidal Salmonella (iNTS) infection—is clearly laid out, and the approach of examining diversity of Salmonella isolates from the environment, animals, and humans living in households of index iNTS cases is appropriate to address it. 

A strength of the approach is extensive environmental sampling, which supports their conclusions that household environmental sources harbor diverse NTS strains but are not major sources of iNTS. A limitation of the design (which the authors discuss) is that in was not feasible to examine community, food, and/or water sources. Another unknown is timing of infections given the sampling delay of up to 2 weeks following index case enrollment, and it’s possible that there were additional sources of infection/matched isolates that were missed.

In addition, there are several areas where more detail is needed on the study population and sample collection:

- How was household membership defined?

- It would be helpful to include a table summarizing characteristics of the recruited participants, including age group, sex, household size, symptoms in index cases, timing between index enrollment and household sampling, and any other pertinent information from the socio-demographic questionnaire.

- The authors collect 3 consecutive stool samples from humans, 1 from animals, and use environmental “boot-sock” sampling. Recovery rate of Salmonella isolates is similar between human, animal, and environmental sources in aggregate, but I’m unsure if this alone indicates that sampling was sufficient. Are there other measures or control sequences that could be used to compare the level of sampling between households and sources?

- Line 256. “Sample positivity was…similar across different household categories.” In Table 1, positivity rates vary from 0 to 28% between different categories—what do the authors mean in this sentence?

- Line 257. Animal sampling was from 59% of households. Did the rest of the households not have animals, or not perform animal sampling?

Reviewer #3: Study would have benefited from enrolling additional households (cases and controls). But statistical comparisons were not made between groups so small sample size not a concern from a statistical perspective.

**Results**

-Does the analysis presented match the analysis plan?

-Are the results clearly and completely presented?

-Are the figures (Tables, Images) of sufficient quality for clarity?

Reviewer #1: - Please include a flow chart/table to clarify enrollment, yield and characteristics of index cases and households: specify total patients eligible (blood culture sampled), NTS and S. Typhi blood culture positivity, enrolled households, household size, number of samples per household member, age of household members (sampled). If possible, additional data on household members would be interesting to provide more insights on fecal carriage: median interval between sampling and clinical presentation/fever start, presence of fever/gastro-intestinal symptoms in household members and antibiotic treatment. 

- Line 256: Not clear what you mean with "similar accros different household categories", because the yield varies from 10-34%

- Line 259 - 261: "Salmonella bacteria ... household samples". Suggestion to integrate these sentences in first part of paragraph describing overall yield.

- Table 1: Please provide data on the proportion of household members from whom NTS were isolated, if possible stratified according to age.

- Table 2: Please specify that serotypes were inferred from WGS in the table header.

- Table 2: Suggestion to add to the table caption that the table only contains information on Salmonella "resuscitated from the freezer"

- Table 2: Last column is same as first column.

- Line 280 - 285: "There were two instances .... from their stools (Figure 1). Suggestion to rephrase, not clear that first and second sentence are about the same 2 pairs. 

- Figure 1: What does the dark red color refer to? 

- Line 305: ST325 is not mentioned in Table 2 or Figure 1

- Line 308 -310: "Some STs isolated ... human or animal isolates.": Already described in Table 1 and previous paragraph, isn't it? 

- Line 336: "This indicates a very marked geographical diversity between households". I don't fully understand what you mean and don't see how the data shown allow statements on geographical diversity. See comment above on providing a map with geographic distribution of sampled households and isolates. 

- Line 361: The paragraph on AMR is a bit out of the scope and methodologically relatively weak due to the absence of phenotypic testing. If kept in the manuscript, I strongly suggest to be more specific: which molecular mechanisms were assessed but not present, provide numbers of isolates in which each mechanism was retrieved (instead of "all or some of the isolates") and add for each mechanism the antibiotic (class) to which it confers resistance. Is the presence of fosA7 gene clinically/epidemiologically relevant?

Reviewer #2: Overall, the tables and figures are clear and match the analysis plan. Two areas for improvement:

- In Figure 1, it would be helpful to additionally visualize the household similarity among NTS not linked to invasive disease (results in paragraph 327-338). For example, delineating households where environmental, animal and/or healthy human samples match using purple, green, or dashed lines (as was done in black and red for the matched invasive disease isolates). 

- The individual-level isolate data provided in Table 2 is great, but it’s also hard to digest all this information at once. It would be helpful to have an additional table summarizing the serogroups by source of infection. For example: for each sequence type/serovar/serogroup (in rows) what number & percent were isolated from each household sample type (in columns).

Reviewer #3: (No Response)

**Conclusions**

-Are the conclusions supported by the data presented?

-Are the limitations of analysis clearly described?

-Do the authors discuss how these data can be helpful to advance our understanding of the topic under study?

-Is public health relevance addressed?

Reviewer #1: Abstract: 

- Line 48: "healthy humans were the source of iNTS infections in the household". The study results support indeed the hypothesis of human to human transmission, albeit only based on 2 pairs. This association however does not allow statements on the source of infection/direction of transmission. I also suggest to integrate the hypothesis on a common source of infection in the abstract conclusion. 

- Line 51 - 53: In addition to the absence of Salmonella Typhi isolated from the environment, the zero positivity from human faeces is remarkable and might indicate limited sensitivity for isolation of iNTS from faecal samples too. 

Introduction: 

- The introduction is long and its readability and storyline can be improved. Please restructure the paragraphs, so that each paragraph focusses on one topic. 

- Information on NTS in HICs vs typhoid fever vs iNTS is a bit scattered across the paragraphs. To help the reader, it might help to focus on main concepts, e.g. "NTS in HICs = zoonotic transmission, intestinal infection" vs. "typhoid fever = human-to-human transmission, invasive disease" vs "iNTS= transmission not fully elucidated yet but human-to-human transmission hypothesized, invasive disease". 

- Available evidence on iNTS transmission is discussed in the introduction and discussion section. Suggestion to keep in-depth discussion of transmission for the discussion section and only briefly refer to possible transmission routes in introduction. When discussing available evidence, please group evidence as much as possible by hypothesis, e.g. all data in favor of human-to-human transmission together. It is a bit hard to follow now due to jumping back and forth between isolates from animals, healthy children and the environment.

Discussion: 

- The discussion would benefit from restructuring, e.g. according to the recommendations of Docherty and Smith: BMJ 1999;318:1224-5 (summary of findings; comparison with previous literature ; strengths and limitations ; meaning of the study and understanding possible mechanism; implications for practice, policy and future research).

- Suggestion to shorten the summary of the evidence in the introduction and move the details on current evidence on NTS transmission to the discussion, so you can discuss all evidence at once here and compare your data with it.

- Line 412 - 415: "While we failed ... in this area". Do you think the technical challenges with isolation of S. Typhi might have affected the yield of NTS too? If so, do you think it might have caused a bias towards less iNTS isolation compared to non-invasive NTS? It would be interesting to know how many stool samples you finally obtained per household member, as you expect incremental yield if multiple stool samples are analysed. 

- Line 420: Is it possible to assess the hypothesis on ST313 Lineage 3 based on your WGS data?

Reviewer #2: The conclusions that this study provides evidence for household transmission of iNTS and no evidence for animal or environmental household reservoirs are supported. 

The evidence for person-to-person transmission comes from only 2 of the 28 index iNTS cases, and not all potential sources including food, water, and community sources could be sampled (which the authors discuss in the limitations). So, the conclusion on lines 374-375 that “Our findings support the hypothesis that humans are the primary reservoir for iNTS in sub-Saharan Africa” is not strongly supported, and the authors may consider rephrasing. 

In addition, the paper would be strengthened by:

- Including the limitation of potential delays between infection/exposure and sampling.

- Discussing what is needed in future studies to test the hypothesis that healthy humans were the source of iNTS infections in the household (e.g., prospective sampling) and what the results of such a study would mean for iNTS prevention and control.

Reviewer #3: (No Response)

**Editorial and Data Presentation Modifications?**

Reviewer #1: Minor modifications: 

- Line 78: suggestion to harmonize typhoidal and non-typhoidal (both without capital)

- Line 97-99: "In Sub-Saharan Africa (SSA), however, the epidemiological picture... carries an estimated case-fatality of 14.7%." Split sentence to improve readability.

- Line 101-104: "Two Salmonella serotypes ... ST11 being the most frequently identified." Split sentence to improve readability. Furthermore the list with countries from which data on the serotype distribution are available is incomplete. To avoid the suggestion that enteritidis and typhimurium are not the main serotypes in other, non-listed sub-Saharan African countries, either complete the list or use more general wording.

- Line 305: Suggestion to replace "associated with different households" by "sampled in different households"

- Line 329 - 334 "Striking examples ... in the same households". Hard to read, please split in multiple sentences and rephrase. 

- Line 364: "52 genomes" sounds a bit strange to me

- Line 365: Please harmonize, either addition of s to abbreviation for plural or not ("9 ST313s, 3 ST3257")

- Line 397 - 399: "The lack of overlap ... very striking". Please correct sentence.

- Line 403-406: "The presence of the same STs... from iNTS cases". Please split sentences and rephrase.

Reviewer #2: The paper is mostly very well written. Some comments:

- Please make sure to define all acronyms the first time they are presented in the text. 

- Suggest including study location (Blantyre, Malawi) and dates (Mar 2015-Oct 2016) in the abstract.

- Suggest replacing “Africa” in the title with “Malawi”.

Reviewer #3: (No Response)

**Summary and General Comments**

Reviewer #1: The study provides additional insights to an important knowledge gap, i.e. transmission of iNTS in SSA. It strenghtens the hypothesis that non-typhoidal Salmonella in sub-Saharan Africa have adapted to the human host and human-to-human transmission. 

Major comments (see section-specific comments) on the study and manuscript are : 

- The manuscript is relatively long, contains redundancies and might benefit from restructuring. Some sentences should be rephrased to improve clarity. Avoid discussion of the data in the results section. Review by one of the native English speaking co-authors is required. 

- A flow chart clarifying patient and household enrollment and sampling should be added, including a brief sociodemographic description of cases and households.

- A map should be added to display the geographical distribution of households.

Reviewer #2: This is an important paper that provides evidence for household transmission of invasive Salmonella infections in Blantyre, Malawi, and no evidence for household environmental or animal reservoirs. This is consistent with previous work in Kenya, Burkina Faso, and The Gambia, and could support a future study that investigates Salmonella household transmission prospectively. As described above, there are several areas where additional details are needed, including:

1) Further detail on sample collection and population in a new table and in the methods.

2) Further detail on isolate diversity/similarity by household (Figure 2) and sampling sources (new table corresponding current Table 2).

3) Given the limitations of this study, discussion of what additional information/studies are needed to inform iNTS mitigation.

Reviewer #3: (No Response)

PLOS authors have the option to publish the peer review history of their article (what does this mean?). If published, this will include your full peer review and any attached files.

Reviewer #1: No

Reviewer #2: Yes: Kirsten E Wiens

Reviewer #3: No
---

## [Decision Letter · Decision Letter 1]

18 Oct 2022

Dear Dr Ashton,

Thank you very much for submitting your manuscript "Case-control investigation of invasive Salmonella disease in Malawi reveals no evidence of environmental or animal transmission of invasive strains, and supports human to human transmission." for consideration at PLOS Neglected Tropical Diseases. As with all papers reviewed by the journal, your manuscript was reviewed by members of the editorial board and by several independent reviewers. The reviewers appreciated the attention to an important topic. Based on the reviews, we are likely to accept this manuscript for publication, providing that you modify the manuscript according to the review recommendations. 

Sincerely,

Andrew S. Azman

Section Editor

Reviewer's Responses to Questions

**Key Review Criteria Required for Acceptance?**

**Methods**

-Are the objectives of the study clearly articulated with a clear testable hypothesis stated?

-Is the study design appropriate to address the stated objectives?

-Is the population clearly described and appropriate for the hypothesis being tested?

-Is the sample size sufficient to ensure adequate power to address the hypothesis being tested?

-Were correct statistical analysis used to support conclusions?

-Are there concerns about ethical or regulatory requirements being met?

Reviewer #1: No remaining pertinent comments on the methods in the revised version

Reviewer #2: - The definition for household membership was provided in response to my comment, but not added to the methods. Please add this definition to the methods.

- I now understand what the authors mean in the sentence “Sample positivity was…similar across different household categories” based on the authors' response, but the text in the manuscript is still unclear. I suggest editing this sentence to clarify what “household categories” refers to and why having comparable sample positivity is important.

Reviewer #3: With these revisions, this study and manuscript appear to satisfy these criteria. The overarching study goal/hypothesis is clear, and the study design and population appropriate. Explicit statistical hypothesis testing was not undertaken, so sample size is not a major concern. No ethical concerns.

**Results**

-Does the analysis presented match the analysis plan?

-Are the results clearly and completely presented?

-Are the figures (Tables, Images) of sufficient quality for clarity?

Reviewer #1: • Table 1: Please add a legend explaining the abbreviations: NA is not applicable or not available? 

• Figure 1: I suggest the authors to reconsider how the data can be better visualized to be more informative.

o Part A: If feasible, I suggest harmonization with categories used in table 2: e.g. distinguish between index (typhoid fever), index (iNTS) and control households by using small pie charts with size adjusted to number of households per area. If not feasible, numbers stating the numbers of households might be more informative than the bubbles that are now used.

o Part B: It is very hard for the reader to grasp the data here, since the information is scattered around the map instead of geographically visualized. The times 2 or 3 to indicate multiple isolates from the same sequence type is a bit lost amidst all the labels. Please reconsider how you can visualize this more clearly without losing the granularity on source and ST. Maybe working with multiple chloropleth maps (one per source and one for invasive + household member) might be a solution. 

• Line 321-322 (legend figure 2): “Isolates of the same ST from different niches within the same household are indicated by purple, green and magenta outlines.” It is not clear to me how I should interpret this. Can you please explain in more detail to facilitate a better understanding?

• Line 338: I suggest the authors to specify that serotypes mentioned in the following paragraphs were determined in silico.

Reviewer #2: (No Response)

Reviewer #3: This version of the manuscript is much clearer than the previous version. Necessary tables and figures are provided and labeled appropriately. The analysis appears to match the original plan.

**Conclusions**

-Are the conclusions supported by the data presented?

-Are the limitations of analysis clearly described?

-Do the authors discuss how these data can be helpful to advance our understanding of the topic under study?

-Is public health relevance addressed?

Reviewer #1: • Line 450 – 452: The high positivity rate on household level in typhoid fever index- and control households compared to iNTS index- and control households is an interesting finding and largely resulted from a difference in yield of animal and environmental samples. Do the authors have any hypothesis/potential explanation for this finding? And what about the absence of Salmonella in fecal samples of iNTS-control households? 

• Limitations: 

o The authors now explicitly state that none of the household members was treated with antibiotics for iNTS disease, so that no household members had to be excluded for this reason. However, if I understood it correctly, it is possible that some household members were on antibiotics at the moment of sampling. If so, I recommend mentioning this as a limitation.

o The authors explained in their rebuttal letter that it was not feasible to collect data on stool frequency/consistency of household members or to know how many stool motions were sampled per household member. Since this might impact the yield, I recommend adding this to the limitation section.

Reviewer #2: - Suggest rephrasing lines 75-77 to something like: “Our findings support a hypothesis that invasive Salmonella infections are transmitted within households from non-animal or environmental sources.”

Reviewer #3: The strengths, weaknesses, and limitations of available data are thoroughly discussed. Conclusions are well aligned with results and don't over-generalize. Findings are well situated in our existing knowledge of the topic and clearly describe what this study adds to our understanding. Authors have also done a good job of highlighting what understanding this study design is not able to support (ie cross sectional so not able to definitely show transmission pathways).

**Editorial and Data Presentation Modifications?**

Reviewer #1: (No Response)

Reviewer #2: - Suggest removing “, and supports human to human transmission” from the title.

- The “tracked changes” version of the manuscript does not correspond to the clean version of the manuscript provided. Please upload a manuscript that shows what changes were made to arrive at the revised manuscript.

Reviewer #3: (No Response)

**Summary and General Comments**

Reviewer #1: o The manuscript has much improved after restructuring and addition/modification of tables/figures. Congratulations to the authors for their thorough revision! 

o There are still some typos or syntax errors due to the textual modifications. I advise the authors to scan the manuscript one more time to correct these.

Reviewer #2: I thank the authors for their extensive revisions and detailed responses. My remaining comments are minor and refer to areas of the text where further clarity would be helpful.

Reviewer #3: Overall edits are very good. This updated version of the manuscript is much clearer and persuasive while being explicit about what conclusions are and are not supported by the findings. The Author Summary is excellent.

PLOS authors have the option to publish the peer review history of their article (what does this mean?). If published, this will include your full peer review and any attached files.

Reviewer #1: No

Reviewer #2: Yes: Kirsten E. Wiens

Reviewer #3: No

Figure Files:

Data Requirements:

Reproducibility:

References

---

## [Editor Report · Decision Letter 2]

23 Nov 2022

Dear Dr Ashton,

We are pleased to inform you that your manuscript 'Case-control investigation of invasive Salmonella disease in Malawi reveals no evidence of environmental or animal transmission of invasive strains, and supports human to human transmission.' has been provisionally accepted for publication in PLOS Neglected Tropical Diseases.

Best regards,

Andrew S. Azman

Section Editor

---

## [Editor Report · Acceptance letter]

7 Dec 2022

Dear Dr Ashton,

We are delighted to inform you that your manuscript, "Case-control investigation of invasive Salmonella disease in Malawi reveals no evidence of environmental or animal transmission of invasive strains, and supports human to human transmission.," has been formally accepted for publication in PLOS Neglected Tropical Diseases.

Best regards,

Shaden Kamhawi

co-Editor-in-Chief

Paul Brindley

co-Editor-in-Chief
